# Autophagy, Apoptosis, the Unfolded Protein Response, and Lung Function in Idiopathic Pulmonary Fibrosis

**DOI:** 10.3390/cells10071642

**Published:** 2021-06-30

**Authors:** Pawan Sharma, Javad Alizadeh, Maya Juarez, Afshin Samali, Andrew J. Halayko, Nicholas J. Kenyon, Saeid Ghavami, Amir A. Zeki

**Affiliations:** 1Center for Translational Medicine, Division of Pulmonary, Allergy and Critical Care Medicine, Jane & Leonard Korman Respiratory Institute, Sidney Kimmel Medical College, Thomas Jefferson University, Philadelphia, PA 19107, USA; pawan.sharma@jefferson.edu; 2Department of Human Anatomy and Cell Science, Max Rady College of Medicine, Rady Faculty of Health Sciences, University of Manitoba, Winnipeg, MB R3E 3P4, Canada; alizadej@myumanitoba.ca; 3Davis Lung Center, School of Medicine; Division of Pulmonary, Critical Care, and Sleep Medicine, University of California, Davis, CA 95616, USA; mmjuarez@ucdavis.edu (M.J.); njkenyon@ucdavis.edu (N.J.K.); 4Apoptosis Research Centre, School of Natural Sciences, National University of Ireland, H91 W2TY Galway, Ireland; afshin.samali@nuigalway.ie; 5Department of Physiology and Pathophysiology, Max Rady College of Medicine, Rady Faculty of Health Sciences, University of Manitoba, Winnipeg, MB R3E 3P4, Canada; andrew.halayko@umanitoba.ca; 6Veterans Affairs Medical Center, Mather, CA 95655, USA; 7Research Institute of Hematology and Oncology, Cancer Care Manitoba, Winnipeg, MB R3E 0V9, Canada; 8Faculty of Medicine, Katowice School of Technology, 40-555 Katowice, Poland; 9Autophagy Research Center, Shiraz University of Medical Sciences, Shiraz 7134845794, Iran

**Keywords:** unfolded protein response, endoplasmic reticulum stress, autophagy, apoptosis, idiopathic pulmonary fibrosis, lung fibrosis, pulmonary function test, lung function, lung physiology

## Abstract

Autophagy, apoptosis, and the unfolded protein response (UPR) are fundamental biological processes essential for manifold cellular functions in health and disease. Idiopathic pulmonary fibrosis (IPF) is a progressive and lethal pulmonary disorder associated with aging that has limited therapies, reflecting our incomplete understanding. We conducted an observational study linking molecular markers of cell stress response pathways (UPR: BiP, XBP1; apoptosis: cleaved caspase-3; autophagy: LC3β) in lung tissues from IPF patients and correlated the expression of these protein markers to each subject’s lung function measures. We hypothesized that changes in lung tissue expression of apoptosis, autophagy, and UPR markers correlate with lung function deficits in IPF. The cell stress markers BiP, XBP1, LC3β puncta, and cleaved caspase-3 were found to be elevated in IPF lungs compared to non-IPF lungs, and, further, BiP and cleaved caspase-3 co-localized in IPF lungs. Considering lung function independently, we observed that increased XBP1, BiP, and cleaved caspase-3 were each associated with reduced lung function (FEV1, FVC, TLC, RV). However, increased lung tissue expression of LC3β puncta was significantly associated with increased diffusion capacity (DLCO), an indicator of alveolar–capillary membrane function. Similarly, the co-localization of UPR (XBP1, BiP) and autophagy (LC3β puncta) markers was positively correlated with increased lung function (FEV1, FVC, TLC, DLCO). However, the presence of LC3β puncta can indicate either autophagy flux inhibition or activation. While the nature of our observational cross-sectional study design does not allow conclusions regarding causal links between increased expression of these cell stress markers, lung fibrosis, and lung function decline, it does provide some insights that are hypothesis-generating and suggests that within the milieu of active UPR, changes in autophagy flux may play an important role in determining lung function. Further research is necessary to investigate the mechanisms linking UPR and autophagy in IPF and how an imbalance in these cell stress pathways can lead to progressive fibrosis and loss of lung function. We conclude by presenting five testable hypotheses that build on the research presented here. Such an understanding could eventually lead to the development of much-needed therapies for IPF.

## 1. Introduction

Idiopathic pulmonary fibrosis (IPF) is the most common of the interstitial lung diseases affecting the older population [1,2]. IPF is a progressive and fatal disorder with a post-diagnosis median survival of 3–5 years [1,2,3]; there are few options in terms of available pharmacological therapies. The etiology of IPF remains unknown. However, the fundamental pathological processes involved in IPF may be triggered by chronic and/or repetitive damage to the alveolar epithelium of the lung [1,3]. The injured epithelium results in the activation of other types of lung cells, namely, alveolar fibroblasts. The resultant wound healing response by the activated fibroblasts leads to the accumulation of extracellular matrix (ECM), causing phenotypic yet irreversible lung fibrosis that leads to loss of lung function [1,2,3]. In addition, aging is increasingly recognized as a very important factor in the development of IPF, where fibroblast senescence can contribute to IPF pathogenesis [4]. 

Macroautophagy (hereafter referred to as “autophagy”), a lysosome-mediated catabolic process that degrades damaged organelles and aggregated proteins, serves as a survival mechanism against cellular stress [5,6,7,8]. Relevant to the fibrotic response in IPF, autophagy also promotes phenotype conversion of myofibroblasts [9] with elevated levels of ECM proteins [10]. The unfolded protein response (UPR) is an indicator of endoplasmic reticulum (ER) stress, and autophagy is a mechanism by which ER stress may be resolved in the cell; however, sustained ER stress leads to apoptosis mediated by the activation of caspase proteases [11]. Apoptosis dysregulation is important in the pathogenesis of pulmonary fibrosis and can contribute to disease progression [12], and it has also been implicated in alveolar epithelial cell pathology in individuals with lung fibrosis exposed to thoracic irradiation [13].

UPR markers are detected in fibroblasts and lung tissues of IPF patients, indicating a relationship between UPR and IPF [14,15]. Although markers of UPR are elevated in fibroblasts from IPF patients, autophagy markers are lower in these cells compared to fibroblasts from non-IPF patients [16]. Others have also reported impaired autophagy flux in IPF lungs [17,18]. However, despite the lower autophagy activity, both autophagy and UPR appear to be necessary for TGFβ1-induced pro-fibrotic response in IPF fibroblasts [16]. This illustrates the complex interactions among the fundamental cell stress responses necessary to regulate lung physiological function. 

Cross-talk between these mechanisms has been proposed [19,20,21,22,23], where unresolved UPR can also induce apoptosis [24], a possible cause of the severe scarring observed in IPF lung tissues. To our knowledge, we are the first to examine the direct association of these three cell stress response mechanisms simultaneously (including UPR, autophagy, and apoptosis) with lung function measures in patients with IPF. In this pilot clinical observational and hypothesis-generating study, we test whether changes in lung tissue expression of apoptosis, autophagy, and UPR markers correlate with the loss of lung function in individuals with IPF. Our results highlight the physiological importance of cell stress signaling pathways in IPF, which has led us to several new testable hypotheses. The end result of such investigation may lead to much-needed novel therapies targeting cell stress pathways.

## 2. Materials and Methods

### 2.1. Study Subjects 

Patients from the University of California (U.C.) Davis Interstitial Lung Disease (ILD) Clinic (IRB-approved registry, protocol number 295939) with lung-biopsy-confirmed IPF were selected for this study. Diagnosis of IPF was based on the criteria outlined in American Thoracic Society (ATS) guidelines [3], including independent confirmation by a pulmonary pathologist, and all other causes of interstitial lung diseases were excluded based on a thorough clinical evaluation. Inclusion criteria for initial entry into the ILD clinic registry were as follows: (i) under the care of a UC Davis Internal Medicine Pulmonary Clinic pulmonologist, (ii) a diagnosis of ILD, (iii) 18 years of age or older. We obtained de-identified human lung tissues from deceased IPF patients from the period 2005 to 2012 (Table 1). De-identified lung function data obtained at the U.C. Davis Pulmonary Function Lab are also included in Table 1.

The comparative non-IPF lung tissues were obtained from patients undergoing lung resection surgery for lung cancer in 2010 at the University of Manitoba, Department of Thoracic Surgery (approved by the Human Research Ethics Board, protocol HS14752 (H2002:150)). These non-IPF lung tissue slides were of peripheral lung specimens from tumor-free, non-involved tissues, independently confirmed by a pulmonary pathologist. Patient characteristics and lung function data are also provided in Table 2.

### 2.2. Immunofluorescence Immunohistochemistry 

H&E staining was performed using 4 μm sections, while immunostaining was performed using primary antibodies for LC3β (Proteintech 18725-1-AP), cleaved caspase-3 (CST #9661), BiP (ab21685, Abcam), and XBP1 (ab37152, Abcam). Alexa-Fluor-conjugated antibodies (Molecular Probes) and DAPI were used to stain the nuclei. Images were captured and analyzed in a blinded fashion using a Nikon confocal microscope, keeping both laser intensity and detector sensitivity settings constant, and analyzed using ImageJ software [25]. For LC3β, we measured the puncta as representative of autophagosomes that confirm autophagy flux.

### 2.3. Selection of Markers for Autophagy, Apoptosis, and the Unfolded Protein Response

We have chosen LC3β as one of the key markers of autophagosome formation based on published literature [26,27] and our own findings [28,29,30]. Quantitatively, the measurement of autophagosomes is based on the numbers of puncta stained with LC3β. Increased LC3β puncta do not necessarily demonstrate autophagy flux but indicate the exact numbers of autophagosomes without differentiation between formation and degradation. GRP78 (BiP) is an ER chaperon that is responsible for the docking of the three UPR sensors (IRE1, ATF6, PERK). We have previously shown that immunohistochemical staining for BiP is a reliable marker for the activation of UPR (16, 30). Hence, we used this approach in our current work. Additionally, we also used XBP/sXBP to monitor the activation of the IRE arm of UPR signaling. XBP/sXBP plays an important role in organ fibrosis, including that of the lung [16] and the gut [31]. Further, we chose cleaved caspase-3 as an apoptosis marker as it is the key caspase for cell death [32,33]; we have demonstrated this approach in our previous study [30]. Of note, we did not use cleaved PARP-1 as an apoptosis marker since PARP-1 is a marker for DNA repair mechanisms via PARylation and may, therefore, interfere with the repair mechanism(s) in IPF [34].

### 2.4. Quantification of Immunofluorescence Intensity and Colocalization

We have calculated the immunofluorescence intensity and co-localization of different markers using Image Zen software, as described previously [28]. Briefly, we created a uniform region of interest (ROI) for each marker and quantified green or red fluorescence intensities. Considering the total area of each ROI uniform, we calculated green/red fluoresce intensity and were able to generate a percent value for some of the comparisons. For measuring green and red co-localization in each ROI, we quantified the co-localization using the merged fluorescence histograms created by Image Zen and by measuring the intensity of yellow fluorescence in the ROI.

### 2.5. Pulmonary Function Tests

Complete pulmonary function testing (PFT) was performed at the U.C. Davis PFT Lab according to ATS/ERS Guidelines. PFT included spirometry, flow-volume loops, lung volumes (measured by body plethysmography), and diffusion capacity. Lung function was measured serially both before and after the diagnosis of IPF, including pre- and post-bronchodilator measures. For our linear correlation analyses between cell stress markers and lung function, PFT data within 1–1.5 years of the date of the lung tissue biopsy (i.e., pathological diagnosis) was used to avoid increasing time-based biases. More specifically, these PFT data (over 1–1.5 years) straddled the date of lung biopsy, where the date of biopsy was situated in the middle of this time range. For a chronic progressive disease such as IPF, changes in lung function are typically slow and gradual and are, therefore, unlikely to change significantly over the observed limited duration, thus reducing the added variability that may accompany longer time ranges. With this approach, we were able to justify our linear correlation analyses between cell stress protein markers (single time point based on the date of lung biopsy) and PFT data (multiple time points over 1–1.5 years). This understanding then allows us to approximate lung function measures similar to a single time point measure.

### 2.6. Statistical Analysis

Linear regression was used to assess correlations between patient-matched lung function parameters (absolute values for forced expiratory volume in the first second (FEV1), forced vital capacity (FVC), residual volume (RV), total lung capacity (TLC), diffusion capacity of carbon monoxide (DLCO)), and IPF lung tissue cell stress markers (BiP, XBP1, LC3β puncta, cleaved caspase-3). Of note, the correlation plots do not all contain an equal number of data points, which simply reflects randomly missing PFT values. We placed greater interpretative power on the absolute values of lung function (e.g., as measured in liters for FVC or TLC) rather than the percent-predicted values (e.g., FVC % predicted) that are normalized to patient population norms. The absolute values for lung function are direct measures of pulmonary physiology without the potentially confounding effects of normalized values across larger population norms that do not necessarily reflect our rather small, single-center cohort (Table 1). 

Given that this was a proof-of-principle study focused on discovery and hypothesis generation, we had a small sample size (four IPF lungs), and, thus, limited statistical power. Therefore, for all linear correlation analyses involving lung function data, we used a *p*-value cut-off of <0.1 to indicate statistical significance unless otherwise stated. Confidence intervals are included in the figures, shown by the dotted curved lines flanking the regression line. Goodness of fit was determined using R^2^ values. Of note, we only show figures from those lung function analyses that are statistically significant. Non-significant statistical correlations are not shown unless deemed relevant.

For analyses involving comparisons between cell stress markers by immunofluorescence, a *p*-value of <0.05 was chosen to indicate statistical significance (with two-way alpha). The choice of this lower (and more commonly used) *p*-value cut-off reflects high enough statistical power at the available sample size. We used Student’s t-test for all statistical comparisons comprising parametric data. GraphPad Prism version 9.0.0 was used to conduct all analyses.

## 3. Results

### 3.1. Co-Localization of Apoptosis, Autophagy, and UPR Markers in IPF Lung Tissues

Using confocal microscopy, we measured differences in the markers of apoptosis, autophagy, and UPR in comparable regions of lung tissue from IPF patients and non-IPF patients without lung fibrosis. We observed a significantly higher co-localization of UPR (BiP) and apoptosis (cleaved caspase-3) markers (*p* < 0.001) in lung samples from IPF subjects, including some involvement of small airways (Figure 1A,B). Conversely, the UPR marker BiP did not co-localize with the autophagosome marker (LC3β puncta) in any of the tissues we assessed, including lung parenchyma and small airways (Figure 1C,D). While the UPR maker XBP1 did co-localize with LC3β puncta (light orange color) in both IPF and non-IPF lung samples, this was not significantly different between the patient and control groups (Figure 1E,F). The tissue sections evaluated included lung parenchyma and the small or terminal airways. 

These results indicate that ER stress/UPR and apoptosis are concomitantly induced in IPF lung cells. These correlative data are in agreement with our recent mechanistic studies that demonstrated spliced-XBP1 as a driving factor for collagen production in IPF fibroblasts (16). Interestingly, despite the non-colocalization of BiP with LC3β puncta in IPF lungs and insignificant co-localization of XBP1 with LC3β puncta in both IPF and non-IPF lungs, the total labeling of BiP, XBP1, LC3β puncta, and cleaved caspase-3 as independent measures was significantly elevated in IPF lungs compared to non-IPF lungs (Figure 1G–J). 

We did not use cell-specific markers to co-stain epithelial cells, fibroblasts, endothelial cells, or immune cells along with UPR, apoptosis, and autophagy markers. However, based on the immunofluorescence staining in Figure 1, we observe that epithelial cells, sub-epithelial cells (which likely include fibroblasts), and lung parenchymal tissues were involved.

### 3.2. Correlation between Cell Stress Markers and Lung Function in IPF

Our IPF lung tissue sections were obtained from IPF patients confirmed to have restrictive physiology by PFT and biopsy-proven usual interstitial pneumonia (UIP), the classical histopathology seen in IPF (Table 1). By comparison, our non-IPF patients served as negative controls, as described in Table 2. FEV1 signifies changes in the volume expired in the first second of a forced breath and is typically reduced in any significant lung pathology that involves the airways and/or lung parenchyma. This value has significant prognostic power in terms of overall patient health and correlates with mortality. Reductions in lung volumes (FVC, RV, TLC) in IPF represents the presence of restrictive physiology due to lung fibrosis. The loss in FVC and TLC has predictive power in interstitial lung diseases, including IPF. The DLCO represents alveolar–capillary membrane function, which is also typically reduced in IPF due to the presence of lung parenchymal fibrosis. 

We examined the associations between cell stress marker expression and lung function parameters and found that greater expression (i.e., labeling by immunofluorescence) of both XBP1% and BiP% (UPR) (Figure 2A,B) and cleaved caspase-3% (apoptosis) (Figure 2C) in IPF lungs was associated with reduced FEV1, FVC, TLC, and RV, respectively. Conversely, greater expression of LC3β puncta% was positively correlated with increased DLCO (Figure 2D). This suggests that both increased UPR and increased apoptosis are independently associated with reduced lung function in IPF, as measured by FEV1, FVC, TLC, and RV, while the presence of autophagosomes in IPF lungs is associated with higher DLCO. These are important lung function parameters that also relate to the functional capacity of IPF patients.

We then assessed correlations between co-localized UPR, autophagy, and apoptosis markers. We also hypothesized that co-localized markers of UPR and autophagy in IPF lung tissues would correlate with lower lung function values, similar to our XBP1, BiP, and cleaved caspase-3 results. However, we found that the opposite was true. Higher immunofluorescence of overlapping markers XBP1 and LC3β puncta was consistently associated with significantly greater FEV1, FVC, and TLC (Figure 3A,B). Higher immunofluorescence of overlapping markers BiP and LC3β puncta was associated with significantly increased DLCO (*p* = 0.0067) (Figure 3C). However, co-localized LC3β puncta and BiP were not significantly correlated with any lung function parameters (*p* > 0.1) (data not shown). There were no statistically significant correlations between overlapping markers of BiP and cleaved caspase-3 for any lung function parameter (data not shown). These results suggest that co-localized markers of UPR (XBP1, BiP) and autophagy (LC3β puncta) are associated with improved lung function (increased FEV1, FVC, TLC, DLCO). Figure 4 graphically summarizes our results and describes testable hypotheses to direct future research in this area. 

We also conducted additional analyses to corroborate our linear correlation results. As described in greater detail in the Appendix A, we re-analyzed a subset of the data shown in Figure 2 and Figure 3. After randomly matching cell stress marker expression with lung function data, we repeated linear correlations. The results obtained were similar to the original analyses in Figure 2 and Figure 3. Notably, on repeat analysis after random allocations, the XBP1% and FEV1 negative correlation was stronger and became statistically significant (*p* = 0.084, R^2^ = 0.33) (Appendix A).

## 4. Discussion

Taken together, our results suggest that UPR, autophagy, and apoptosis protein expression markers are significantly increased in IPF compared to non-IPF lungs. Moreover, in IPF lungs, we observed significantly increased co-localization of BiP and cleaved caspase-3 compared to non-IPF lungs (Figure 1A), which points toward a co-active UPR and apoptotic signaling pathway in IPF. These novel findings of increased UPR and apoptosis markers in IPF lungs also correlated with changes in lung function. We discovered that UPR (XBP1, BiP) and apoptosis (cleaved caspase-3) were each associated with a reduction in FEV1, FVC, TLC, and RV. In contrast, increased expression of either LC3β puncta or co-localized LC3β puncta and UPR markers (XBP1, BiP) were associated with increased DLCO or increased FEV1, FVC, TLC, and DLCO, respectively. The co-existence of these cell stress pathways in IPF pathogenesis and the link to changes in lung function (a key clinical determinant) are unique findings. 

Cellular phenotype is tightly regulated via autophagy and UPR in lung epithelial and mesenchymal cells [35,36]. For example, TGFβ1 has been shown to induce changes in autophagic flux (decreased LC3β puncta expression) and increased cellular senescence in IPF lung fibroblasts [10,37]. Autophagy induction also controls UPR-induced senescence in bronchial epithelial cells, while autophagy inhibition blocks this [7]. We also know that nintedanib (a multiple tyrosine kinase inhibitor) downregulates ECM production and promotes autophagy in IPF fibroblasts while inhibiting TGFβ1 signaling, which confirms the role of autophagy in the regulation of IPF fibroblasts (synthetic phenotype) [38]. Conversely, we previously reported that TGFβ1-induced ECM protein production was significantly higher in IPF lung fibroblasts compared with non-IPF donors [16]. While TGFβ1 concomitantly induced autophagy and profibrotic signaling, leading to the accumulation of ECM proteins in vitro, autophagy-related signaling was significantly lower in lung fibroblasts from IPF subjects compared to non-IPF fibroblasts. Further, inhibition of autophagy signaling prevented TGFβ1-induced ECM production by lung fibroblasts obtained from both non-IPF and IPF donors [16]. Interestingly, TGFβ1 induced UPR markers only in IPF lung fibroblasts, suggesting a greater-than-anticipated role of ER stress and UPR signaling in lung fibrosis.

Our results bring a level of nuance not always present in in vitro or animal in vivo studies because we relate the complex existence of cell stress protein markers in human IPF lungs to lung function for those same subjects. Putting our previous results into context, we learned that while there is a significant increase in the accumulation of autophagosomes (LC3β puncta) in IPF lungs compared to control non-IPF lung tissues (which shows dysregulation of autophagy flux in IPF lungs) (Figure 1I), the presence of LC3β puncta alone (Figure 2D), or when LC3β puncta are co-localized with UPR markers (Figure 3), is associated with *improved* lung function. Of note, the presence of LC3β puncta can represent either inhibition or increase in “autophagy flux” (Figure 4C). Our previous studies indicate that (i) autophagy is necessary for the fibrotic response in IPF fibroblasts, and (ii) autophagy inhibition reduces this fibrotic response in lung fibroblasts [16]. Combining these two observations would suggest that in human IPF, there is endogenous inhibition of autophagy through mechanisms that may be reliant on changes in cellular ADP/ATP ratios or a potential cross-talk between apoptosis and autophagy (Bcl2-Beclin-1) [38,39,40,41] that, in turn, leads to reduced fibrosis and thus, improved (or better preserved) lung function (Figure 4C). This is an interesting and new hypothesis that can be tested, which could shed new insights into the role of autophagy in IPF.

We believe such mechanisms could be controlling the cellular phenotype (epithelial-to-mesenchymal transition), cellular secretome, and cellular responses to apoptosis, which favors the downregulation of pro-fibrotic signaling to prevent further deterioration of lung function in IPF [42]. Despite the observation that alveolar epithelial cells in IPF appear to be damaged or injured with high rates of apoptosis, the actual causative factor(s) are not entirely known. Evidence is now emerging to support the association between alveolar epithelial cell apoptosis and lung fibrosis. Although the mechanisms that cause alveolar epithelial cell death are not completely understood, it is known that TGFβ1 plays an important role as a potent inducer of apoptosis in alveolar epithelial cells, predominantly by modulating Fas-mediated apoptosis via caspase-8 activation and downregulation of p21 [43]. It has also emerged that TGFβ1 can activate p38/MAPK and early growth response gene (Egr)-1 in a SMAD-dependent and SMAD-independent manner, leading to the induction of pro-apoptotic signaling molecules such as caspase-3 [44,45]. Further, while inducing apoptosis, TGFβ1 promotes the imbalance of Blc family members by stimulating Bax and Bid in the murine lung. TGFβ1–mediated SMAD3 signaling is required for the upregulation of death-associated protein kinase, which is essential for TGFβ1–induced apoptosis of lung epithelial cells. Collectively, TGFβ1 can induce both canonical and non-canonical signaling to promote alveolar epithelial cell apoptosis and lung fibrosis. 

Our data support the newly emerging concept in the field that lung function changes in IPF are unlikely to be attributed to a single perturbed cell stress marker in isolation but are more likely the result of a complex network of cell stress events leading to disease. One alternative interpretation of our results is that the combination of increased UPR and autophagy is necessary for maintaining homeostatic lung function in IPF. However, we cannot invoke causation given our study design; therefore, additional research is warranted to further dissect the underlying mechanisms and their impact on lung function at different stages of disease and among various types of IPF. 

Our correlation analysis between cell stress markers and lung function in IPF subjects suggests a more complex interaction. We previously reported that the UPR is an important driver of TGFβ1-induced collagen production in primary human lung IPF fibroblasts (16). However, to our knowledge, no one has directly linked measures of UPR markers with lung function in patients with IPF. We found that both increased UPR markers XBP1 and BiP are associated with decreased lung function (FEV1, FVC, TLC) (Figure 2A,B). Interestingly, these correlations are the exact opposite when UPR is co-localized with the autophagy marker LC3β puncta (Figure 3), thus highlighting the expected complexity inherent to these cell stress signaling pathways. BiP is the upstream chaperon that inhibits PERK (protein kinase R (PKR)-like endoplasmic reticulum kinase), IRE1 (inositol-requiring enzyme 1), and ATF6 (activating transcription factor 6) chaperons via direct interaction in non-stressed conditions [20,46,47]. Therefore, BiP expression is representative of the overall activation of UPR and the potential involvement of all three arms of this pathway [48]. IRE1 has both kinase and RNAse activity, and its RNAse activity is responsible for the splicing of XBP1 into “spliced XPB1” (XBP1s) [49]. XBP1s is a transcription factor that is responsible for cell proliferation, endoplasmic reticulum (ER)-associated degradation (ERAD), and lipid biosynthesis [20]. On the other hand, UPR is also connected to autophagy flux via PERK and the IRE1 arm through the regulation of LC3β expression. UPR can also regulate apoptosis through modulation of Bcl2 family proteins and CHOP expression [20]. Therefore, we can conclude that changes in BiP expression can potentially be representative of the general regulation of UPR while changes in XBP1 expression only represent the IRE1 arm. Further investigation into how these pathways interact to impact lung cell physiology and, in turn, lung function, symptoms, and patients’ functional capacity is warranted.

The transcription factor XBP1s is important in fibrosis, as shown in several previous experimental models [50,51]. XBP1s is responsible for inflammatory-mediated peritoneal fibrosis in peritoneal dialysis, and the IRE1 endonuclease inhibitor STF083010 inhibits this effect [52]. On the other hand, XBP1 regulates liver fibrosis via the transport and Golgi organization-1 (TANGO1)-collagen axis [53] and the regulation of ATG7-dependent autophagy in hepatic stellate cells [54]. In addition, IRE1 inhibitor 4μ8C blocks liver and skin fibrosis via the IRE1–XBP1 axis in mice [55]. Our recent report also showed that targeting the IRE1 RNAse component inhibits TGFβ1-induced ECM deposition in IPF fibroblasts [16]. Taken together, XBP1 appears to be an important component of UPR in the regulation of fibrosis in multiple organs, including the lung.

Except for an isolated negative correlation with RV (Figure 2C), cleaved caspase-3, when co-localized with BiP, did not reveal any significant correlations with other lung function measures. While other studies have reported apoptosis activation in airway epithelial cells in human IPF lungs [56,57], which is associated with insufficient and dysregulated repair mechanisms, our results suggest that increased apoptosis in IPF lungs may not play a major role in determining lung function (at least with respect to the most relevant PFT measures pertinent to IPF, i.e., FEV1, FVC, TLC, DLCO). In addition, few in vitro and in vivo studies show that dysregulated apoptosis in myofibroblasts leads to lung tissue fibrosis, including in mouse models of TGFβ-induced pulmonary fibrosis [58]. Conversely, nintedanib, an FDA-approved anti-fibrotic drug for the treatment of IPF, which delays the decline in lung function (i.e., preserves FVC), can induce apoptosis in lung fibroblasts, leading to decreased proliferation and reduced production of ECM proteins such as collagen [59,60]. One explanation for this discrepancy could be differences in experimental models (human tissue vs. cells/vs. mouse models of lung fibrosis). In addition, the complexity of how these cell stress pathways converge in human IPF, which has evolved over many years, provides an ongoing challenge to our understanding. 

Our study has significant limitations, including the small sample size and reduced statistical power, as well as the observational study design that does not permit us to determine causation. While our design allows for hypothesis generation, the small sample size precludes any generalizability to IPF or other forms of lung fibrosis. However, our results highlight the potentially critical role of cellular ER stress/UPR, apoptosis, and autophagy as convergent pathways and their impact on lung function and, therefore, their potential impact on symptoms and IPF disease progression. Further studies are needed to determine whether inhibition of UPR and/or inhibition or induction of autophagy pathways can preserve lung function in patients with IPF. Another potential limitation of our study design is the domain of lung function measures collected over a focused time period (1–1.5 years), which allowed correlations to cell stress markers measured at a single time point (i.e., date of lung biopsy). We address this above in the Pulmonary Function Tests and Statistical Analysis sections; however, we cannot fully exclude potentially hidden biases with unknown confounders. However, our independent analysis, provided in the Appendix A, addresses one potential confounding effect. This additional analysis showed that our original allocations and analysis were valid in terms of the linear correlation outcomes and directions of these correlations between cell stress markers and lung function parameters. 

In summary, there appears to be an intriguing paradox or disconnect between the observed increases in cell stress markers in IPF lung tissues and the functional consequences of this increased expression with respect to lung function. While single markers in isolation correlate with ***reduced*** lung function (XBP1, BiP, cleaved caspase-3) when viewed in conjunction with autophagy, we observed the opposite correlation of ***improved*** lung function (Figure 4). We speculate that the dynamic cross-talk occurring between different lung cells in IPF that simultaneously run all three cell stress programs yields functional consequences, i.e., changes in lung function, that cannot be predicted by simply investigating these cell stress pathways in static lung tissues or as single markers in isolation. As a result, we propose five independent hypotheses worthy of further testing, as indicated in Figure 4. As a further caveat, all three cell stress pathways must be investigated concomitantly in any experimental system to properly understand their dynamic interactions. This is the least we can do regarding these critical cell stress/cell fate pathways to gain the depth of understanding required to impact the fibrosis field. This idea and approach also has clinical relevance because there are inhibitors of UPR proteins as well as both inhibitors and inducers of autophagy [61,62], both of which (or in combination) may prove to be useful in the future treatment of IPF. 

## Figures and Tables

**Figure 1 cells-10-01642-f001:**
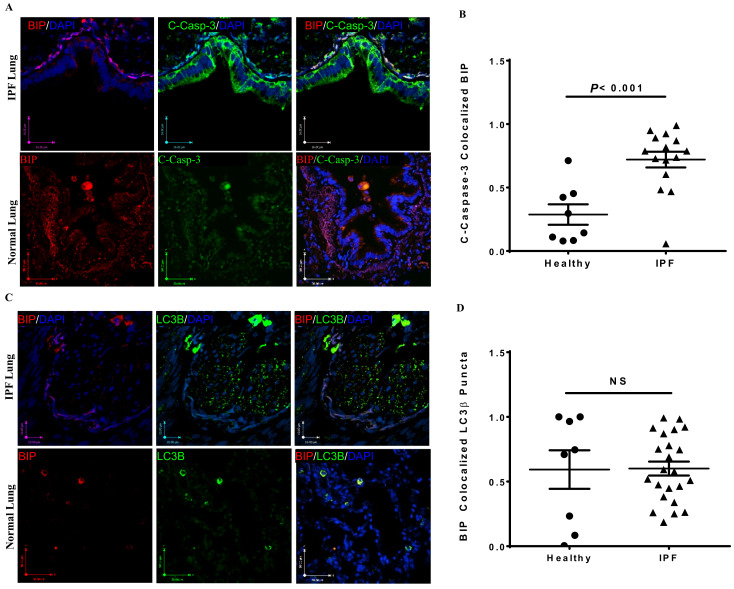
Co-localization of cell stress markers in IPF lung tissues. (**A**) Representative histological images of immunofluorescence confocal microscopy of IPF and non-IPF lung tissues for cleaved caspase-3 (green), BiP (red), and DAPI (blue). Co-localization can be seen as a light orange color in dotted areas in the merged images (far right side). (**B**) IPF lungs had significantly greater co-localization of c-caspase-3 and BiP compared to non-IPF lungs (*p* < 0.001). (**C**) Representative histological images of immunofluorescence confocal microscopy of non-IPF and IPF lung tissues for LC3β puncta (green), BiP (red), and DAPI (blue). Very little co-localization is observed in the merged images (far right side) for both non-IPF and IPF lung tissues. (**D**) There was no statistically significant difference in co-localized BiP and LC3β puncta between IPF and non-IPF lungs (*p* = NS). (**E**) Representative histological images of immunofluorescence confocal microscopy of non-IPF and IPF lung tissues for LC3β puncta (green), XBP1 (red), and DAPI (blue). Some co-localization is observed in the merged images (far right side) for both non-IPF and IPF lung tissues. (**F**) There was no statistically significant difference in co-localized XBP1 and LC3β puncta between IPF and non-IPF lungs (*p* = NS). The percentage of positive signals for BiP (**G**), XBP1 (**H**), LC3β puncta (**I**), and cleaved caspase-3 (**J**) was measured as the ratio of areas between the region of interest and the total image area. This was performed on all the images captured from sections for each lung tissue for all subjects. Data were collected in triplicate, and results are represented as the overall average for all the captured images. **Note:** Magnification was ×40 for images in Panels **A**, **C**, and **E** (scale bars are included in each image). For statistical calculations in Panels **A**–**F**, areas with co-localization of any two cell stress markers were calculated using ImageJ in both non-IPF and IPF lungs. The middle horizontal line in Panels **B**, **D**, **F** represents the median value in each group. Results in Panels **G**–**J** are represented as means ± SD. A *p*-value of <0.05 was considered statistically significant. GraphPad Prism 9 was used for all analyses. Abbreviations: NS (not significant).

**Figure 2 cells-10-01642-f002:**
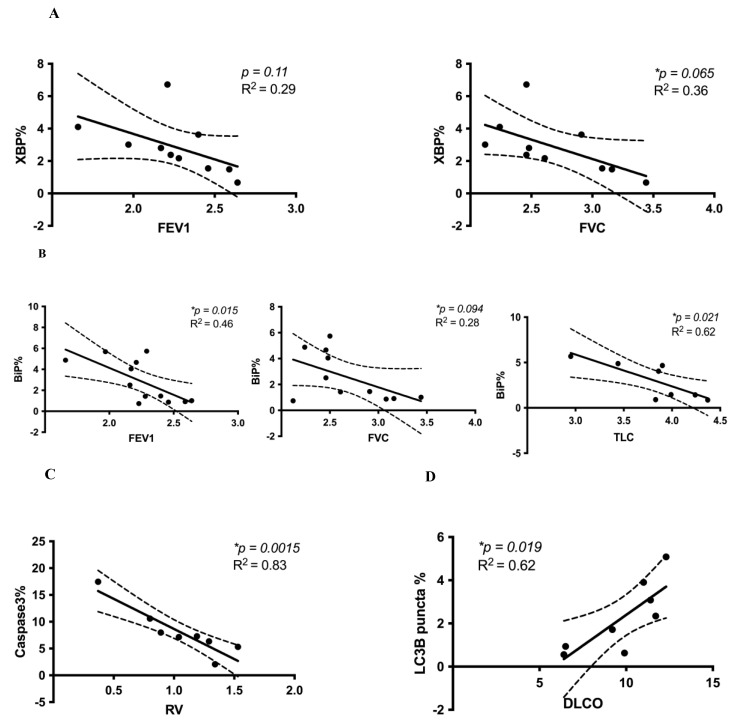
Correlations between cell stress markers and lung function in IPF. (**A**) The UPR marker XBP1% was negatively correlated with FEV1 and FVC. There was a trend of negative correlation between XBP1% and FEV1 (*p* = 0.11, R^2^ = 0.29) and a statistically significant negative correlation between XBP1% and FVC (*p* = 0.065, R^2^ = 0.36). (**B**) The UPR marker BiP% was negatively correlated with FEV1 (*p* = 0.015, R^2^ = 0.46), FVC (*p* = 0.094, R^2^ = 0.28), and TLC (* *p* = 0.021, R^2^ = 0.62). (**C**) The apoptosis marker cleaved caspase-3% was also negatively correlated with RV (*p* = 0.0015, R^2^ = 0.83). (**D**) The autophagy marker LC3β puncta% was positively correlated with DLCO (*p* = 0.019, R^2^ = 0.62). Confidence intervals are included, shown by the dotted curved lines flanking the regression line. Linear regression analyses were performed using GraphPad Prism 9. Lung volumes (FEV1, FVC, RV, TLC) were measured in liters, and the diffusion capacity of carbon monoxide (DLCO) was measured in mL/mmHg/min. * reprents the significant difference.

**Figure 3 cells-10-01642-f003:**
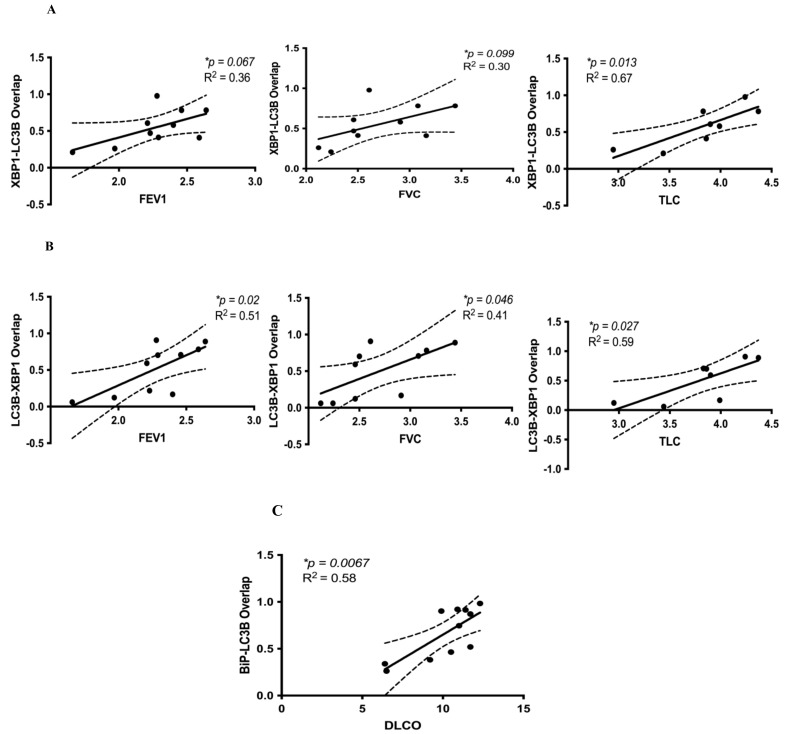
Correlations between co-localized UPR (XBP1, BiP) and autophagy (LC3β puncta) markers and lung function in IPF. (**A**) Co-localized XBP1 and LC3β puncta were positively correlated with FEV1 (* *p* = 0.067, R^2^ = 0.36), FVC (* *p* = 0.099, R^2^ = 0.30), and TLC (* *p* = 0.013, R^2^ = 0.67). (**B**) Co-localized LC3β puncta and XBP1 were positively correlated with FEV1 (* *p* = 0.02, R^2^ = 0.51), FVC (* *p* = 0.046, R^2^ = 0.41), and TLC (**p* = 0.027, R^2^ = 0.59). (**C**) Co-localized BiP and LC3β puncta were positively correlated with DLCO (* *p* = 0.0067, R^2^ = 0.58). Confidence intervals are included, shown by the dotted curved lines flanking the regression line. Linear regression analyses were performed using GraphPad Prism 9. Lung volumes (FEV1, FVC, TLC) were measured in liters, and the diffusion capacity of carbon monoxide (DLCO) was measured in mL/mmHg/min.

**Figure 4 cells-10-01642-f004:**
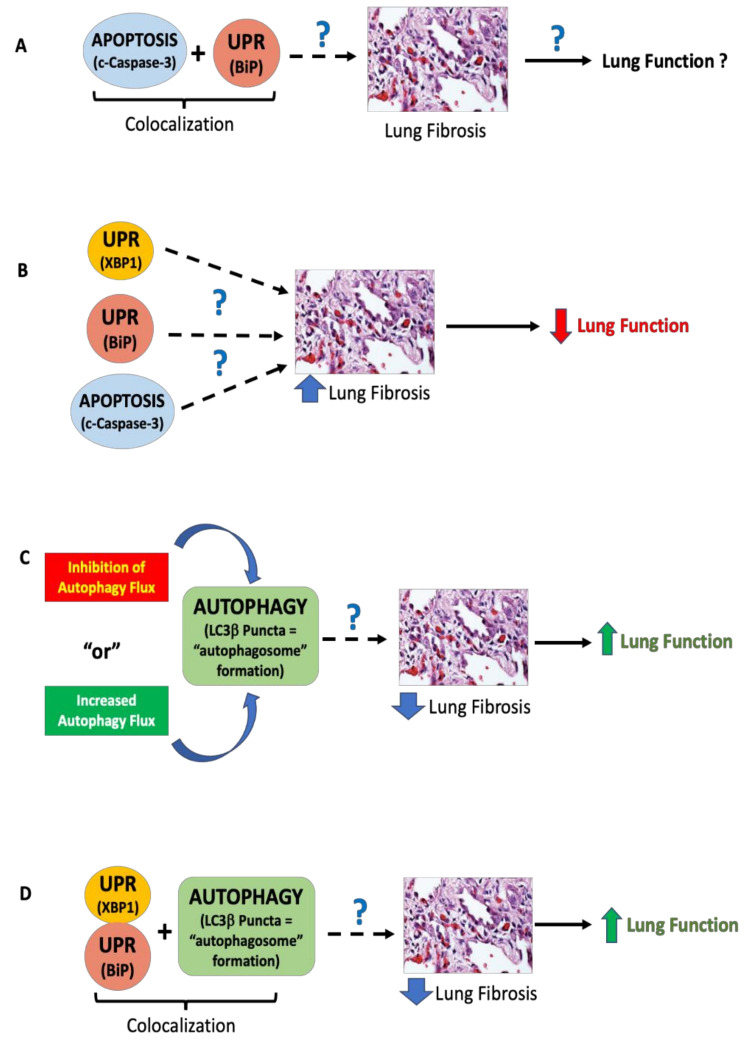
Five testable hypotheses regarding the relative contributions of UPR, apoptosis, and autophagy to pulmonary fibrosis and lung function changes. The confocal and linear correlation data presented in Figure 1, Figure 2 and Figure 3 highlight critical connections linking cell stress biology, lung fibrosis, and the resultant lung function changes in patients that lead to debilitating symptoms. (**A**) Co-localization of cleaved caspase-3 and BiP was significantly greater in IPF lungs than non-IPF lungs, but the causal link to lung fibrosis and lung function remains unknown. (**B**) UPR markers XBP1 and BiP and apoptosis marker cleaved caspase-3 were independently associated with lower lung function in IPF, but the cross-talk between these pathways remains incompletely understood. (**C**) The increase in autophagosome formation (i.e., increased expression of LC3β puncta) could represent either active autophagy flux or the inhibition of autophagy flux. Our data show that autophagy is definitely involved and correlates with improvement in the lung function parameter (i.e., improvement in DLCO); however, the directionality of autophagy remains unknown. (**D**) Co-localization of UPR (XBP1, BiP) and autophagy (LC3β puncta) was associated with higher lung function; however, the mechanism of this observation is not yet known. Our data suggest that within the context of active UPR in IPF lungs, the presence of autophagy changes the consequence of UPR and leads to higher lung function. Whether this occurs via the inhibition or activation of autophagy flux remains unknown. The results from Figure 1 and Figure 2, depicted graphically in Panels **A** and **B**, lead to **Hypothesis 1.** *Cross-talk and dysregulation between apoptosis and UPR pathways cause lung fibrosis, which leads to progressive loss of lung function.* The results from Figure 2 and Figure 3, depicted graphically in Panels **C** and **D**, lead to the following four hypotheses: **Hypothesis 2a.** *Autophagy inhibition reduces lung fibrosis, which, in turn, preserves or improves lung function.* **Hypothesis 2b.** *Autophagy activation reduces lung fibrosis, which, in turn, preserves or improves lung function.* **Hypothesis 3a.** *Within the context of active UPR (XPB1 and BiP), the inhibition of autophagy reduces lung fibrosis, which improves lung function.* **Hypothesis 3b.** *Within the context of active UPR (XPB1 and BiP), the activation of autophagy reduces lung fibrosis, which improves lung function.* Note: Dotted line arrows indicate multiple intervening steps not shown. Solid line arrows indicate a direct effect of lung fibrosis on lung function.

**Table 1 cells-10-01642-t001:** IPF Patient Demographics and Characteristics.

	ID1	ID2	ID3	ID4	ID1-4 ^$^
Age (deceased, yrs)	63	59	85	72	69.8 ± 11.5
Sex	M	M	F	M	
Ethnicity	White	White	White	Hispanic	
**Lung Function** ^$^					
FEV1 (L)	2.52 ± 0.11	2.12 ± 0.14	1.77 ± 0.10	2.16 ± 0.09	2.17 ± 0.22
FEV1%p	71.8 ± 2.6%	56 ± 3.6%	95.3 ± 3.8%	74.8 ± 2.9%	74.5 ± 10.1%
FVC (L)	3.15 ± 0.22	2.33 ± 0.18	2.39 ± 0.14	2.51 ± 0.08	2.58 ± 0.28
FVC%p	69.8 ± 4.6%	46.7 ± 3.2%	93.3 ± 5.0%	67.4 ± 2.6%	68.4 ± 11.9%
FEV1/FVC	80%	90.1%	74.1%	86.1%	84.1%
RV (L)	0.77 ± 0.35	0.77 ± 0.05	1.54 ± 0.5	1.31 ± 0.19	1.10 ± 0.42
RV%p	34 ± 15.9%	34 ± 2.8%	65 ± 24.0%	57.8 ± 7.7%	48.3 ± 17.9%
TLC (L)	4.06 ± 0.28	3.14 ± 0.26	3.92 ± 0.68	3.91 ± 0.25	3.81 ± 0.45
TLC%p	59.7 ± 4.04%	43.5 ± 3.5%	79.5 ± 14.9%	62.8 ± 2.8%	61.5 ± 12.7%
DLCO	8.5 ± 1.8	6.9 ± 0.71	12.9 ± 2.7	10.5 ± 1.5	9.9 ± 2.3
DLCO%p	31 ± 6.2%	22.5 ± 2.12%	66.5 ± 13.4%	45.9 ± 5.9%	42.8 ± 14.1%
Lung Pathology ^#^	UIP	UIP	UIP	UIP	-
Clinical Diagnosis	IPF	IPF	IPF	IPF	-
Comorbidities	AF, CAD, DM, CKD	HIV/AIDS, BPH, SCC (rectal)	HTN, AV dis, MV dis., RBBB, TAH	GERD, HPL, PVD, AF, Divrt, TIA	

^$^ noted as mean ± SD. ^#^ Based on wedge lung biopsy. *Abbrev.* M (male), F (female), L (liters), %p (percent predicted), FEV1 (forced expiratory volume in the first second), FVC (forced vital capacity), RV (residual volume), TLC (total lung capacity), DLCO (diffusion capacity of carbon monoxide), UIP (usual interstitial pneumonitis), IPF (idiopathic pulmonary fibrosis), AF (atrial fibrillation), CAD (coronary artery disease), DM (diabetes mellitus), CKD (chronic kidney disease), HIV/AID (human immunodeficiency virus/acquired immunodeficiency syndrome), BPH (benign prostatic hypertrophy), SCC (squamous cell carcinoma), HTN (hypertension), AV (aortic valve), MV (mitral valve), dis. (disease), RBBB (right bundle branch block), TAH (total abdominal hysterectomy), GERD (gastroesophageal reflux disease), HPL (hyperlipidemia), PVD (peripheral vascular disease), Divrt. (diverticulosis), TIA (transient ischemic attack).

**Table 2 cells-10-01642-t002:** Non-IPF Patient Demographics and Characteristics.

	ID1	ID2	ID3	ID1-3 ^$^
Age (yrs)	76	79	73	76 ± 3
Sex	M	M	M	-
Smoking Status (pack-years)	60 *	30 ^@^	45 ^@^	
Lung Function ^$^				
FEV1 (L)	1.86	2.28	3.07	2.40 ± 0.61
FEV1%p	65%	78%	99%	80.67 ± 17.2%
FVC (L)	2.85	3.34	3.71	3.3 ± 0.43
FEV1/FVC	65%	68%	83%	72 ± 9.6%
Diagnosis based on PFT	Moderate Obstructive Disease	Mild Obstructive Disease	Normal	
COPD GOLD				
Classification	Stage II	Stage II	N/A	
Tumor	Lung Cancer	Lung Cancer	Lung Cancer	

^$^ noted as mean ± SD. * Current smoker; ^@^ Ex-smoker; *Abbrev.* M (male), F (female), L (liters), %p (percent predicted), FEV1 (forced expiratory volume in the first second), FVC (forced vital capacity), COPD (chronic obstructive pulmonary disease), GOLD (Global Initiative for Chronic Obstructive Lung Disease), PFT (pulmonary function test), N/A (not applicable).

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
