# Peer review of "Autophagy, Apoptosis, the Unfolded Protein Response, and Lung Function in Idiopathic Pulmonary Fibrosis"

_cells, 2021, doi:10.3390/cells10071642_

Round 1

Reviewer 1 Report

In the present study, the author selected one of markers for autophagy, apoptosis, and unfolded protein response and investigated their correlations with lung function in IPF patients. Given the fact that the autophagy, apoptosis, and unfolded protein response are critical biological processes in the pathogenesis of IPF, the study is interesting. However, the sample size is very small, the study is underpowered. Specifically:

  1. Autophagy, apoptosis, and unfolded protein response represent three different but critical biological processes with a complicated network for IPF, it is oversimplified to see their correlations based on the co-IF staining for two arbitrarily selected markers. Furthermore, it is not clear about the rationale why these markers were specifically selected to represent Autophagy, apoptosis, and unfolded protein response, respectively.
  2. The sample size is really small for these heterogeneous populations, 4 for IPF, 3 for non-IPF. For IPF patients, there are three male, one female, three white but one Hispanic. For 3 patients with non-IPF, they are all male with lung cancer. Thus, it is questionable whether those are truly controls. Thus, any comparisons between IPF vs non-IPF may not make any sense. Any differences for lung function between IPF vs non-IPF? 
  3. In figure, the dot in each figure is not clearly introduced, from one patient or two slides from one patients, since there are only four IPF patients in total. Also, it is not clear about the Y-axis in several figures. How were these data generated for the co-localized markers in figure 1 and overlapped in figure 3?
  4. In fact, many questions (?) in Figure 4 for their five Testable Hypotheses make a good sense since there are indeed questionable because of  the small sample size but for a complicated biological processes of diseases.  

Author Response

Author Responses to Reviewer Critiques

We thank reviewers for their constructive feedback which has helped us to improve the quality of this manuscript. We appreciate the opportunity to provide detailed and thoughtful responses, as indicated below, in a good faith attempt to improve our report overall. Reviewers’ numbered comments (C) in italics are followed by our point-by-point responses (R). Changes to the text in the revised ‘MARKED’ manuscript are highlighted in red.

Reviewer #1

Comments: In the present study, the author selected one of markers for autophagy, apoptosis, and unfolded protein response and investigated their correlations with lung function in IPF patients. Given the fact that the autophagy, apoptosis, and unfolded protein response are critical biological processes in the pathogenesis of IPF, the study is interesting. However, the sample size is very small, the study is underpowered. Specifically:

C1. Autophagy, apoptosis, and unfolded protein response represent three different but critical biological processes with a complicated network for IPF, it is oversimplified to see their correlations based on the co-IF staining for two arbitrarily selected markers. Furthermore, it is not clear about the rationale why these markers were specifically selected to represent Autophagy, apoptosis, and unfolded protein response, respectively.

R1. We thank the reviewer for this important comment. Although we have included much of this information in the Introduction of the original submission, we are glad to make additional changes as suggested. We have chosen LC3 as a key marker of autophagosome formation based on recommendations from published literature (1, 2), in addition to our own recent publications (3-5). LC3β is a reliable marker of the number of autophagosmes based on the numbers of puncta stained with LC3β. Although it does not show the flux of autophagy, it shows the numbers of autophagosomes without differentiation between formation and degradation, as we have discussed in our original submission. GRP78 (BiP) is an ER chaperon which is responsible for docking of the three UPR arms (IRE1, ATF6, PERK). In our previous work, our team showed that for the BiP protein IHC is a reliable marker for activation of UPR (5, 6), therefore, we used this approach in our current work. We also used XBP/sXBP to monitor for activation of the IRE arm of UPR since XBP/sXBP plays an important role in fibrosis including lung fibrosis (6), Crohn’s disease (7), peritoneal fibrosis (8), cystic fibrosis (9), and hepatic stellate cells. Further, we chose cleaved caspase-3 for apoptosis as it is the final executive caspase for cell death (10, 11), in addition, we have used it in our previous investigations as a reliable IHC marker of apoptosis (5). Of note, we did not use cleaved PARP-1 as an apoptosis marker since PARP-1 can represent DNA repair mechanisms via PARylation and could interfere with the repair mechanisms in IPF samples (12). In short, we hope our esteemed reviewer can appreciate our expertise in the field and that these markers were not selected arbitrarily, but indeed, after considerable deliberation. We have added the above citations to our references where appropriate in the manuscript text. Please see new sub-heading under Materials and Methods called “Selection of Markers for Autophagy, Apoptosis, and Unfolded Protein Response.”

C2. The sample size is really small for these heterogeneous populations, 4 for IPF, 3 for non-IPF. For IPF patients, there are three male, one female, three white but one Hispanic. For 3 patients with non-IPF, they are all male with lung cancer. Thus, it is questionable whether those are truly controls. Thus, any comparisons between IPF vs non-IPF may not make any sense. Any differences for lung function between IPF vs non-IPF? 

R2. We appreciate this important comment by our reviewer. Indeed, this is an important but justifiable limitation in our study design. In doing this study, we did not seek to have sample sizes large enough to provide data that are generalizable to a larger population. If so, then we agree the study design would require a much larger N with proper power and sample size calculations. Rather, this was a proof-of-concept and hypothesis-generating study using as many human samples as we could access at the time of the study. Using these smaller sampling sizes, we reasoned that if there were any correlations they would have to not only be strong, but also statistically significant. The fact we observed statistical significance despite the small sample size could represent a true and strong association. Whether this holds up in larger studies is up to future studies to determine with much larger sample sizes. These three complex biological cell stress pathways (autophagy, UPR, apoptosis) and their involvement in human IPF is a challenge to study in a way that is relevant to the disease process in humans. Our approach is just a beginning to hopefully inspire future studies which can help us identify potential drug targets to treat this devastating illness.

Regarding the three patients with non-IPF lung cancer. We conferred with our co-authors who provided these lungs which were prepared and read by their institutional pathologists, and we were careful to only use specimens that had margins clear of any microscopic evidence of cancer. In addition, we have added Figure 2S in the Supplemental Material section Appendix A which shows both IPF and non-IPF H&E-stained lung histopathology images. As we state in our Materials and Methods section of the original submission:

“These non-IPF lung tissue slides were of peripheral lung specimens from tumor-free, non-involved tissues, confirmed by a pulmonary pathologist.”

Having said this, we do acknowledge this as a potential limitation in our study. However, given that our aforementioned cell stress markers are known to be significantly involved in cancer pathogenesis (1, 3, 13-15), and indeed, did not light up on our confocal images in Figure 1 further shows that these negative controls are acceptable to use for our purposes.

In terms of the differences in lung function between IPF and non-IPF subjects, this information is present in the original Tables 1 and 2. For IPF the mean FEV1%p was 74.5%, and mean FVC was 2.58 L (mean FVC%p 68.4%); and for non-IPF the mean FEV1%p was 80.6%, and mean FVC was 3.3 L (FVC%p not available). TLC and DLCO data were not available for non-IPF subjects. In general, the lung function for IPF subjects trended lower than the non-IPF cohort, however, this is an expected pattern given the disease process we are evaluating. We went a step further and performed a statistical analysis comparing the lung function (FEV1%p and FVC) from both groups using t test, and found there were no statistically significant differences (p>0.05) in the lung function between the two groups.

C3. In figure, the dot in each figure is not clearly introduced, from one patient or two slides from one patients, since there are only four IPF patients in total. Also, it is not clear about the Y-axis in several figures. How were these data generated for the co-localized markers in figure 1 and overlapped in fig 3?

R3. Thank you for this very important clarification. In Figure 1 (B, D, F), for each subject we used multiple slides for each marker to generate the data seen in each Figure (represented as dots on the graphs), and then blindly quantified them. The co-localization has been performed by Zeiss microscope software. We manually identified a region of interest (ROI) and the measured co-localization of green and red based on our recent publication (3) (Figure A, see below).  In Figure 1 (G, H, I, J), the intensity of each marker was calculated using ROI relative to the total area to get the percent value (we have added this detail in the Materials and Methods section).

In Figure 2, the Y-axes represent the percent intensity for each marker which was calculated using ROI to the total area. The X-axes represent the lung function parameters in units measured as described in the Figure legend. The reason that there may be different number of dots across the different panels is that for some measures or some patients, lung function (PFT) data were missing. We only included lung function data that were available and that matched the restricted time-range we defined in our Materials and Methods of 1-1.5 years within the date of lung biopsy for consistency across study subjects. We describe this further under subject heading “Pulmonary Function Tests.”

In Figure 3, the same rationale as above applies as well. In terms of the ‘overlapping (co-localization)’ cell stress markers, these were determined as follows: The co-localization has been performed by Zeiss microscope software. We manually identified region of interest (ROI) and the measure co-localization of green and red based on our recent publication (3) (Figure A, see below).   

C4. In fact, many questions (?) in Figure 4 for their five Testable Hypotheses make a good sense since there are indeed questionable because of the small sample size but for a complicated biological processes of diseases.  

R4. We appreciate this comment from our reviewer. We may not fully understand the comment, please forgive us, but we interpret this comment as being supportive of the proposed hypotheses. Indeed, also supporting the fact that the complexity of these three major cell stress pathways within the context of IPF is not easy to tease out. Again, we make the argument that had there not been any statistically significant correlations in our Results, then we could not have provided this report, nor the confidence of proposing testable hypotheses to lead us to the next step in research. Indeed, we already acknowledged the small sample size, but as we described above under C2/R2, our goal was not the generalizability of our findings, but rather, to discover novel correlations that could inspire new and hopefully relevant and testable hypotheses.  

Additional References

  1. Klionsky DJ, Abdel-Aziz AK, Abdelfatah S, Abdellatif M, Abdoli A, Abel S, et al. Guidelines for the use and interpretation of assays for monitoring autophagy (4th edition)(1). Autophagy. 2021;17(1):1-382.
  2. Klionsky DJ, Abdelmohsen K, Abe A, Abedin MJ, Abeliovich H, Acevedo Arozena A, et al. Guidelines for the use and interpretation of assays for monitoring autophagy (3rd edition). Autophagy. 2016;12(1):1-222.
  3. Shojaei S, Koleini N, Samiei E, Aghaei M, Cole LK, Alizadeh J, et al. Simvastatin increases temozolomide-induced cell death by targeting the fusion of autophagosomes and lysosomes. FEBS J. 2020;287(5):1005-34.
  4. Alizadeh J, Shojaei S, Sepanjnia A, Hashemi M, Eftekharpour E, Ghavami S. Simultaneous Detection of Autophagy and Epithelial to Mesenchymal Transition in the Non-small Cell Lung Cancer Cells. Methods Mol Biol. 2019;1854:87-103.
  5. Yeganeh B, Rezaei Moghadam A, Alizadeh J, Wiechec E, Alavian SM, Hashemi M, et al. Hepatitis B and C virus-induced hepatitis: Apoptosis, autophagy, and unfolded protein response. World J Gastroenterol. 2015;21(47):13225-39.
  6. Ghavami S, Yeganeh B, Zeki AA, Shojaei S, Kenyon NJ, Ott S, et al. Autophagy and the unfolded protein response promote profibrotic effects of TGF-β(1) in human lung fibroblasts. Am J Physiol Lung Cell Mol Physiol. 2018;314(3):L493-L504.
  7. Li C, Grider JR, Murthy KS, Bohl J, Rivet E, Wieghard N, et al. Endoplasmic Reticulum Stress in Subepithelial Myofibroblasts Increases the TGF-beta1 Activity That Regulates Fibrosis in Crohn's Disease. Inflamm Bowel Dis. 2020;26(6):809-19.
  8. Liu A, Song Q, Zheng Y, Xu G, Huang C, Sun S, et al. Expression of XBP1s in peritoneal mesothelial cells is critical for inflammation-induced peritoneal fibrosis. Sci Rep. 2019;9(1):19043.
  9. Lara-Reyna S, Scambler T, Holbrook J, Wong C, Jarosz-Griffiths HH, Martinon F, et al. Metabolic Reprograming of Cystic Fibrosis Macrophages via the IRE1alpha Arm of the Unfolded Protein Response Results in Exacerbated Inflammation. Front Immunol. 2019;10:1789.
  10. Ghavami S, Hashemi M, Ande SR, Yeganeh B, Xiao W, Eshraghi M, et al. Apoptosis and cancer: mutations within caspase genes. J Med Genet. 2009;46(8):497-510.
  11. Ghavami S, Eshraghi M, Kadkhoda K, Mutawe MM, Maddika S, Bay GH, et al. Role of BNIP3 in TNF-induced cell death--TNF upregulates BNIP3 expression. Biochim Biophys Acta. 2009;1793(3):546-60.
  12. Hu B, Wu Z, Hergert P, Henke CA, Bitterman PB, Phan SH. Regulation of myofibroblast differentiation by poly(ADP-ribose) polymerase 1. Am J Pathol. 2013;182(1):71-83.
  13. Tavernier Q, Legras A, Didelot A, Normand C, Gibault L, Badoual C, et al. High expression of spliced X-Box Binding Protein 1 in lung tumors is associated with cancer aggressiveness and epithelial-to-mesenchymal transition. Sci Rep. 2020;10(1):10188.
  14. Shojaei S, Suresh M, Klionsky DJ, Labouta HI, Ghavami S. Autophagy and SARS-CoV-2 infection: Apossible smart targeting of the autophagy pathway. Virulence. 2020;11(1):805-10.
  15. Song X, Lee DH, Dilly AK, Lee YS, Choudry HA, Kwon YT, et al. Crosstalk Between Apoptosis and Autophagy Is Regulated by the Arginylated BiP/Beclin-1/p62 Complex. Mol Cancer Res. 2018;16(7):1077-91.

Reviewer 2 Report

The Authors conducted an observational study linking molecular markers of cell stress response pathways (UPR: BiP, XBP1; apoptosiscleaved caspase-3; autophagy: LC3) in lung tissues from IPF patients and correlated expression of these protein markers to each subject’s lung function measuresThey have hypothesized that changes in lung tissue expression of apoptosisautophagy, and UPR markers correlates with lung function deficits in IPF.  

The work is well written and the obtained results have been well used by the same authors to formulate five testable hypotheses regarding the relative contributions of UPR, Apoptosis, and Autophagy to Pulmonary Fibrosis and Lung Function Changes.  

Therefore, in my opinion this work contributes to the learning of the pathophysiological mechanisms that occur in Idiophatic Pulmonary Fibrosis.  

Author Response

Author Responses to Reviewer Critiques

We thank reviewers for their constructive feedback which has helped us to improve the quality of this manuscript. We appreciate the opportunity to provide detailed and thoughtful responses, as indicated below, in a good faith attempt to improve our report overall. Reviewers’ numbered comments (C) in italics are followed by our point-by-point responses (R). Changes to the text in the revised ‘MARKED’ manuscript are highlighted in red.

Comments: The Authors conducted an observational study linking molecular markers of cell stress response pathways (UPR: BiP, XBP1; apoptosis: cleaved caspase-3; autophagy: LC3) in lung tissues from IPF patients and correlated expression of these protein markers to each subject’s lung function measures. They have hypothesized that changes in lung tissue expression of apoptosis, autophagy, and UPR markers correlates with lung function deficits in IPF.  The work is well written, and the obtained results have been well used by the same authors to formulate five testable hypotheses regarding the relative contributions of UPR, Apoptosis, and Autophagy to Pulmonary Fibrosis and Lung Function Changes.   Therefore, in my opinion this work contributes to the learning of the pathophysiological mechanisms that occur in Idiopathic Pulmonary Fibrosis. 

 Response: We thank our reviewer for his/her comments.

Reviewer 3 Report

The paper is well written and interesting. However, there are some moments that authors should pay attention to to improve the manuscript.

  1. The methods stipulate H&E staining, the results of which are not presented in the manuscript. There is only one recurring picture in the graphical abstract,  which does not give an idea of the severity of IPF on the contrary, reducing the understanding of the figure.
  2. Although lung function is an important diagnostic indicator of IPF, evidence of collagen and ECM deposition must be provided. It can be Masson's trichrome staining, Western blotting, or PCR analysis. 
  3. The connection of apoptosis and canonic SMAD signaling pathway in PF should be added to the discussion

Author Response

Author Responses to Reviewer Critiques

We thank reviewers for their constructive feedback which has helped us to improve the quality of this manuscript. We appreciate the opportunity to provide detailed and thoughtful responses, as indicated below, in a good faith attempt to improve our report overall. Reviewers’ numbered comments (C) in italics are followed by our point-by-point responses (R). Changes to the text in the revised ‘MARKED’ manuscript are highlighted in red.

Reviewer #3

Comments: The paper is well written and interesting. However, there are some moments that authors should pay attention to improve the manuscript.

C1. The methods stipulate H&E staining, the results of which are not presented in the manuscript. There is only one recurring picture in the graphical abstract,  which does not give an idea of the severity of IPF on the contrary, reducing the understanding of the figure.

 R1. We appreciate the reviewer for pointing this out. We have now included Figure 2S in the Supplemental Material that shows H&E-stained lung slides from both our IPF and non-IPF subjects. This figure clearly shows the stark differences in pulmonary fibrosis and alveolar space destruction between the two patient cohorts, and we hope, further clarifies the issue regarding IPF pathology and severity.

 C2. Although lung function is an important diagnostic indicator of IPF, evidence of collagen and ECM deposition must be provided. It can be Masson's trichrome staining, Western blotting, or PCR analysis.

R2. We agree with our esteemed reviewer that evidence indicating fibrosis in our IPF subject lung samples is important to show. Indeed, our new Figure 2S clearly shows evidence of lung fibrosis on H&E staining in our representative IPF lung slide with extensive collagen deposition. All other patients similarly showed lung fibrosis consistent with usual interstitial pneumonitis (UIP), as we previously described (see Table 1). Therefore, we respectfully contend that this is clear evidence of lung fibrosis, which was also independently confirmed by a pulmonary pathologist, as indicated in our original submission under Materials and Methods with further clarification in the revised manuscript, as follows:

“Diagnosis of IPF was based on criteria as outlined in the American Thoracic Society (ATS) guidelines (1, 2) including independent confirmation by a pulmonary pathologist, and…”

C3. The connection of apoptosis and canonic SMAD signaling pathway in IPF should be added to the discussion.

 R3. This is a fair point raised by our reviewer. Indeed, this is a very important connection between apoptosis and SMAD in IPF. We have added the following relevant text to the Discussion to address this point:

 “Despite the observation that alveolar epithelial cells in IPF appear to be damaged or injured with high rates of apoptosis, the actual causative factor(s) are not entirely known. Evidence is now emerging to support the association between alveolar epithelial cell apoptosis and lung fibrosis. Although the mechanisms that cause alveolar epithelial cell death are not completely understood, it is known that TGFβ plays an important role as a potent inducer of apoptosis in alveolar epithelial cells predominantly by modulating Fas-mediated apoptosis via caspase-8 activation and down-regulation of p21 (3). It has also emerged that TGFβ can activate p38/MAPK and early growth response gene (Egr)-1 in a SMAD-dependent and SMAD-independent manner leading to induction of pro-apoptotic signaling molecules such as caspase-3 (4, 5). Further, while inducing apoptosis TGF-β promotes the imbalance of Blc family members by stimulating Bax and Bid in the murine lung. TGF-β–mediated SMAD3 signaling is required for up-regulation of death-associated protein kinase which is essential for TGF-β–induced apoptosis of lung epithelial cells. Collectively, TGFβ can induce both canonical and non-canonical signaling to promote alveolar epithelial cell apoptosis and lung fibrosis.”

Additional References

  1. Raghu G, Collard HR, Egan JJ, Martinez FJ, Behr J, Brown KK, et al. An official ATS/ERS/JRS/ALAT statement: idiopathic pulmonary fibrosis: evidence-based guidelines for diagnosis and management. American journal of respiratory and critical care medicine. 2011;183(6):788-824.
  2. Raghu G, Remy-Jardin M, Myers JL, Richeldi L, Ryerson CJ, Lederer DJ, et al. Diagnosis of Idiopathic Pulmonary Fibrosis. An Official ATS/ERS/JRS/ALAT Clinical Practice Guideline. American journal of respiratory and critical care medicine. 2018;198(5):e44-e68.
  3. Siegel PM, Massague J. Cytostatic and apoptotic actions of TGF-beta in homeostasis and cancer. Nat Rev Cancer. 2003;3(11):807-21.
  4. Thannickal VJ, Horowitz JC. Evolving concepts of apoptosis in idiopathic pulmonary fibrosis. Proc Am Thorac Soc. 2006;3(4):350-6.
  5. Lee CG, Cho SJ, Kang MJ, Chapoval SP, Lee PJ, Noble PW, et al. Early growth response gene 1-mediated apoptosis is essential for transforming growth factor beta1-induced pulmonary fibrosis. J Exp Med. 2004;200(3):377-89.

Round 2

Reviewer 1 Report

Thank you for your excellent responses to the concerns.

Reviewer 3 Report

The manuscript was significantly improved. I would suggest to move the new H&E figure from the supplemental materials in the main text.